# Potential Health Effects of Heavy Metals and Carcinogenic Health Risk Estimation of Pb and Cd Contaminated Eggs from a Closed Gold Mine Area in Northern Thailand

**DOI:** 10.3390/foods11182791

**Published:** 2022-09-09

**Authors:** Paweena Aendo, Michel De Garine-Wichatitsky, Rachaneekorn Mingkhwan, Kamonthip Senachai, Pitchaya Santativongchai, Praphaphan Krajanglikit, Phitsanu Tulayakul

**Affiliations:** 1Faculty of Veterinary Medicine, Kasetsart University, Bangkok 10900, Thailand; 2CIRAD, UMR ASTRE, Kasetsart University, Bangkok 10900, Thailand; 3ASTRE, University Montpellier, CIRAD (French Agricultural Research Centre for International Development), INRAE (French National Research Institute for Agriculture, Food and Environment), 34000 Montpellier, France; 4Department of Social and Environmental Medicine, Faculty of Tropical Medicine, Mahidol University, Bangkok 10400, Thailand; 5Phichit Provincial Livestock Office, Phichit 67000, Thailand; 6Bio-Veterinary Science (International Program), Faculty of Veterinary Medicine, Kasetsart University, Bangkok 10900, Thailand; 7Department of Veterinary Public Health, Faculty of Veterinary Medicine, Kasetsart University Kamphaeng Saen Campus, Nakhon Pathom 73140, Thailand; 8Kasetsart University Research and Development Institute, 50 Ngam Wong Wan Rd., Lat Yao, Chatuchak, Bangkok 10900, Thailand

**Keywords:** heavy metals, egg, poultry, carcinogenic risk, gold mine, Thailand

## Abstract

Gold-mining activities have been demonstrated to result in significant environmental pollution by Hg, Pb, and Mn, causing serious concerns regarding the potential threat to the public health of neighboring populations around the world. The present study focused on heavy-metal contamination in the eggs, blood, feed, soil, and drinking water on chicken farms, duck farms, and free-grazing duck farms located in areas <25 km and >25 km away from a gold mine in northern Thailand. In an area <25 km away, Hg, Pb, and Mn concentrations in the eggs of free-grazing ducks were significantly higher than >25 km away (*p* < 0.05). In blood, Hg concentration in free-grazing ducks was also significantly higher than those in an area >25 km away (*p* < 0.05). Furthermore, the Pb concentration in the blood of farm ducks was significantly higher than in an area >25 km away (*p* < 0.05). The concentration of Cd in drinking water on chicken farms was significantly higher for farms located within 25 km of the gold mine (*p* < 0.05). Furthermore, a high correlation was shown between the Pb (r^2^ = 0.84) and Cd (r^2^ = 0.42) found between drinking water and blood in free-grazing ducks in the area <25 km away. Therefore, health risk from heavy-metal contamination was inevitably avoided in free-grazing activity near the gold mine. The incremental lifetime cancer risk (ILCR) in the population of both Pb and Cd exceeded the cancer limit (10^−4^) for all age groups in both areas, which was particularly high in the area <25 km for chicken-egg consumption, especially among people aged 13–18 and 18–35 years old. Based on these findings, long-term surveillance regarding human and animal health risk must be strictly operated through food chains and an appropriate control plan for poultry businesses roaming around the gold mine.

## 1. Introduction

Gold mining is an important source of heavy-metal pollution in ecosystems, with particularly significant impacts on water, soil, and air, and deleterious effects on living organisms [1,2]. As heavy-metal concentrations build up in organisms within the food chain, animals and human inhabitants in the vicinity of gold-mining operations may be exposed to health risks [3]. Gold is among the extracted minerals with the highest socioeconomic value [4]. However, the industrial-scale production of gold generates huge volumes of waste. The extraction and smelting processes disperse large quantities of heavy metals and toxic chemicals into the surrounding environment, contaminating both the water (including groundwater) and soil [5,6].

Heavy metals associated with gold-mining activities include mercury (Hg), lead (Pb), cadmium (Cd), and manganese (Mn) [7,8,9]. Hg contamination was found in the soil, water, and animals such as birds within a radius of 30 km from a contamination source in the USA [10,11]. Hg has been identified as one of the most toxic nonradioactive materials known to man [12]. It is 10 times more toxic to neurons than Pb and may be a cause of Alzheimer’s disease [13]. It poses a risk to miners, with an estimated 10 to 19 million workers exposed in more than 70 countries [14]. Therefore, Hg is an international priority in terms of toxic pollution and is now a major global concern [12,15]. Pb and Cd are also a threat to human health because of their carcinogenic nature [16,17]. According to the International Agency for Research on Cancer (IARC), they fall into Group 2B (Possibly carcinogenic to humans) and Group 1 (Carcinogenic to humans), respectively. Pb (group 2B) and Cd (group 1) are classified as potential carcinogenicity metals, whereas Mn is considered a non-carcinogenic metal [18]. However, overexposure to Mn is also toxic to the brain, and its accumulation has been associated with neurological impairment, disruption of homeostasis in other metals, and neurotoxicity [19,20,21]). Health hazards associated with the consumption of heavy metal-contaminated food products include neuronal damage, cardiovascular disorders, renal injuries, and risk of cancer and diabetes [22,23]. Additionally, the implications of heavy metals with regards to children’s health have been noted to be more severe compared to adults. The harmful consequences of heavy metals on children’s health include mental retardation, neurocognitive disorders, behavioral disorders, respiratory problems, cancer, and cardiovascular diseases [24].

The largest gold-mining area in Thailand is located in the northern provinces of Phitsanulok, Phetchabun, and Pichit. Over 1.8 million ounces of gold and more than 10 million ounces of silver were produced between 2001 and 2016 before mining activity was legally paused [25]. Even after the closure of operations, contamination by heavy metals is persistent in the environment, as they are non-degradable by natural processes, which may cause serious health concerns [26]. Livestock production is significant in Pichit Province, with the total number of chickens and ducks estimated at >760,000 and >650,000 animals in 2020, respectively, including 85% of free-grazing ducks [27]. Nowadays, heavy-metal mining operations are one of the largest sources of environmental pollution in Thailand, creating significant concerns about food safety, particularly in poultry products (meat and eggs) with potential heavy metals residues that can pose health risks to consumers [28,29].

In a previous study [30], we found that the incremental lifetime cancer-risk levels of Pb and Cd in poultry eggs and meat consumption collected from the central and western regions of Thailand were higher than the cancer limit for children and adults, set at 10^−4^, according to international norms [31,32]. Moreover, children in these regions were at risk when consuming contaminated duck eggs, and the risk was higher than in adults by 3.9 times for Pb and Cd [30]. Despite potential risks for public health, there have been few studies of the impacts of gold mining in Thailand [33,34]. Furthermore, no study has attempted to evaluate the carcinogenic risks concerning the consumption of heavy metals through contaminated chicken and duck eggs in gold-mining areas of Thailand. Therefore, the first aim of the present study was to determine and compare the Hg, Pb, Cd, and Mn concentrations in poultry and farm environments of chicken, ducks, and free-grazing ducks located at various distances from a gold mine, potentially as a source of heavy-metal contamination. A secondary aim was to identify the relationship between heavy metals in animals and the environment. Lastly, the study attempted to assess the carcinogenic risk caused by the consumption of poultry eggs near and far away from a gold-mining area, even though it has been closed since 2016.

## 2. Materials and Methods

### 2.1. Study Area and Sample Collection

This study protocol was approved by Kasetsart University’s Institutional Committee for Animal Care and Use according to the guidelines for animal care under the ethical review board of the Office of the National Research Council of Thailand (protocol number ACKU62-VET-045, 2019). Animal and environmental samples were collected from chicken and duck farms in two areas defined according to their distance from the Chatree gold-mine site, Pichit Province (15.9769 N° 106.4429 E°): (i) <25 km away from gold-mining site: 6 chicken farms, 2 duck farms, and 3 free-grazing duck farms; (ii) >25 km away from the gold mine: 6 chicken farms, 3 duck farms, and 3 free-grazing duck farms (Appendix A). The elevation of the study area was 44 to 87 m above sea level. The average annual rainfall is >1000 mm, with an average wind speed of 7.9 km/h (maximum wind speed of 9.3 km/h) [35,36]. All samples were collected 3 times during January, April, and August 2019. A questionnaire survey for descriptive data collection (*n* = 23 farms) provided the following estimates: average number of chickens/farms = 12 farm range (10–150), average number of ducks/farms = 5 farm range (2000–4000), and average number of free-grazing ducks per farm = 6 farm range (2000–10,000). The average age of the animals on the farms at the time of the survey was >5.0 months.

The samples comprised 5 eggs, 5 blood samples, 1 kg of feed, 1 kg of soil, and 1 L of drinking water, which were randomly pool collected for each farm. The total of chickens, ducks, and free-grazing ducks for each egg and blood sample were 60, 25, and 30 animals, respectively, per time in both areas. Blood was sampled from the inner brachial vein at the wing vein using a 23-gauge needle. A volume of 1.5–2 mL of whole blood was collected in a plastic tube containing heparin to prevent clotting, mixed, and placed on ice for transport to the laboratory in Bangkok. Blood samples were frozen at −20 °C and stored until chemical analysis. One kilogram of pooled topsoil (0–20 cm depth) from five different locations across each farm was kept in fresh polyethylene bags in a refrigerator at a temperature below 4.0 °C, then immediately brought to the laboratory [37]. Drinking-water samples were preserved by using 2–3 mL of Conc. HNO_3_ to prevent metal precipitation and then put in a refrigerator at below 4.0 °C until analysis [38].

### 2.2. Analytical Methods

All samples were duplicated for analysis. Egg and feed samples were dried in an oven at 60 °C for 24 h. One gram of sample was digested with 10 mL of 65% HNO_3_ and 2 mL of 30% H_2_O_2_ and heated on a block at 100–120 °C until the sample solution was completely digested (CALA-accredited standard operating procedures [39] and MET-CHEM-ICP-01A (modified from EPA Method 200.8 for biological samples)). A total of 100 µL of each blood sample was mixed with Additive B for Hg analysis according to USEPA 7473, ASTM D-6722-01, and D-7623-10 test methods. For Pb, Cd, and Mn analysis, 100 µL of each blood sample was mixed with ammonium phosphate, Triton X-100, and 18.2 MΩ (Milli-Q) water (Hitachi scientific instrument technical data). The soil samples were air-dried for 24 h at 105 °C, then crushed and sieved over a 2 Diameter sieve. After passing through the sieve, the sample fraction was taken for analysis [40,41]. One gram of soil was digested using 65% HNO_3_, 18.2 MΩ (Milli-Q) water and 30% H_2_O_2_ heated on the block at 100–120 °C [42]. The drinking-water samples were filtrated with filter paper (4 pore size 25 µm Whatman), then mixed with 1:1 (65% HNO_3_: 18.2 MΩ (Milli-Q) water) before analysis [43,44,45]. For Hg analysis, all samples were analyzed using a Mercury analyzer, Model MA-3000 (Nippon, Japan). Pb, Cd, and Mn analysis was carried out using a Graphite Furnace Atomic Absorption Spectrophotometer, Model: ZA 3000 (Hitachi, Japan).

### 2.3. Analytical Procedure

The percent recoveries from standard reference materials (human blood (Seronorm TM trace elements whole blood and Bio-Red lyphockek^®^), control Enviro Mat Ground water, high (ES-H-3) and control Enviro Mat Contaminated Soil (SS-1, SS-2)) of Hg, Pb, Cd, and Mn averaged 99.0 ± 4.68, 101.4 ± 3.39, 102.92 ± 2.26, and 102.06 ± 4.83, respectively. The calibration curves were constructed via linear regression with at least 5 points and were considered optimal if the regression coefficient was ≥0.99. Relative standard deviations of the heavy metals were less than 5%. The analytical detection limits of Hg, Pb, Cd, and Mn were 0.004, 1.01, 0.07, and 10 µL L^−1^, respectively.

### 2.4. Statistical Analysis

Data analysis was performed using GraphPad Prism (version 5.01, 2007 for Windows, GraphPad Software, Inc., San Diego, CA, USA). Normality of variance was tested by the Kolmogorov–Smirnov test. All metal levels were tested and appeared as non-parametric data. The Mann–Whitney U test was performed for comparisons of the concentrations of each metal in eggs, blood, feed, soil, and drinking water between the two groups of farms (areas < 25 km vs. >25 km away). The Pearson correlation coefficient was calculated to estimate the relationship between heavy-metal concentrations in eggs across all samples. A *p*-value of 0.05 was used to determine the significance in all tests.

### 2.5. Carcinogenic Risk Calculation

#### 2.5.1. Estimated Daily Intake

The estimated daily intake (*EDI*) values [32,46] for metals were calculated using:(1)$$EDI=C\times W_{IR}$$
where *C* is the mean concentration of metals (μg L^−1^) and *W_IR_* is the chicken- and duck-egg ingestion rate of Thai people (g kg^−1^d^−1^); 3–6 yo = 26.58, 5.03; 6–13 yo = 29.00, 4.74; 13–18 yo = 29.14, 4.82; 18–35 yo = 28.96, 6.06; 35–65 yo = 22.32, 5.49; and 65 yo up = 18.06, 4.49 [47].

#### 2.5.2. Estimation of Carcinogenic Risk

The estimated incremental probability of an individual developing cancer over a lifetime as a result of potential exposure to carcinogenic heavy metals through egg ingestion was calculated using *ILCR* [48,49] as:(2)ILCR=EDI×CSF

*CSF* is the carcinogenic slope factor (mg kg^−1^d^−1^). The *CSF* value for Pb and Cd was 0.0085 and 0.38 mg kg^−1^d^−1^, respectively [31,50]. Cancer risk surpassing 10^−4^ was unacceptable and considered to pose significant health effects related to cancer [31,32].

## 3. Results and Discussion

### 3.1. Heavy-Metal Concentrations and Correlation among Samples

#### 3.1.1. Area < 25 km from the Gold Mine

On farms located <25 km away from the gold mine, the average and standard variations in concentrations of Hg, Pb, and Mn in the eggs of free-grazing ducks were significantly higher than for farms located >25 km away (*p* < 0.05). Moreover, the Hg concentrations in eggs from both farm ducks and free-grazing ducks were found to be 1.5–3 times higher than the standard limit set by the Ministry of Public Health of Thailand in 2020 (Table 1). In seabirds, Hg contamination has been associated with reduced egg hatchability, possibly via altered egg-turning behavior by parents [51]. Additionally, embryonic exposure to Hg may result in carry-over effects on later chick development [52]. Williams et al. (2017) reported that, after Pb contamination in birds, the weight and length of bird eggs were significantly decreased, whereas lesions to the liver, kidney, spleen, and thymus were increased [53]. In terms of Cd, the LD_50_ of duck and chicken embryos was 8 μg, besides which they experienced a decrease in hatchability and hepatocyte damage [54,55]. The average concentrations of Cd in this study ranged between 8.94 and 13.13 μg kg^−1^, which are also considered harmful levels to poultry in both areas. In blood, the average Hg concentration in free-grazing ducks was also significantly higher for farms located >25 km away from the mine (*p* < 0.05; Table 2).

People living and working in both artisanal and gold-mining areas are frequently exposed to Hg, which is used for gold extraction. It is estimated that about 15 million miners are affected globally [56]. Additionally, exposure to other toxic metals such as arsenic (As), Pb, Cd, and Mn may occur through mining-related activities and could be ingested via air, sediment, water, or food contamination [57,58]. Mining activities such as excavation, crushing, and milling may result in the increased liberation of these toxic metals. Although the gold metal is collected at the end of the mining process, metals may end up in the tailing dumps at mining locations, thus presenting an exposure hazard for people living and working in these mining areas [56]. Santos et al. (2020) found that surface-sediment samples collected in an area under the influence of gold mining were polluted (moderately to seriously) [59]. Wilson et al. (2004) reported that Hg concentrations in blood increased during the breeding season in female birds from Northern Alaska, USA [60]. Pb concentrations in the blood of female birds increased significantly (possibly via re-release of stored lead from bones) during incubation [61]. The degree of contamination in the area depended on where the poultry lived, as well as species, age, sex, size, and time since the pyrite mine was opened. The trophic level influences the accumulation of metal in organs and tissue [62,63]. In our study, the age of farm ducks and free-grazing ducks in an area <25 km away was higher than in an area >25 km away, as shown in Appendix A, which was one of the factors correlated with heavy-metal contamination in poultry <25 km away.

Free-grazing ducks raised in fields are supplied by natural water sources, which may present a high risk of exposure to chemicals in contaminated environments [64]. Similarly, Yabe et al. (2013) reported that free-range chickens raised near a lead–zinc mine in Zambia accumulated greater concentrations of Pb and Cd in the liver than confined broilers [65]. Moreover, Grace and MacFarlane (2016) reported that the concentration of Pb in homegrown eggs in Australia was generally higher than in commercial eggs [66]. In a previous study in Phichit Province, Northern Thailand, we found that Pb and Cd concentrations in the intestines of free-grazing ducks were significantly higher than in those of ducks from intensive farms, whereas Cd concentration in the livers of free-grazing ducks was also higher than in those on intensive duck farms [67]. This study indicated that free-grazing ducks were a health risk and contamination risk due to their exposure to Hg, Pb, and Mn within 25-km areas, making it imperative to avoid grazing near gold-mine sites.

**Table 1 foods-11-02791-t001:** Mean ± SD, the median, minimum, and maximum values of Hg, Pb, Cd, and Mn concentrations in poultry egg (µgkg^−1^ dry weight).

Metals		Chicken		Duck Farm		Free-Grazing Duck		# Chicken Egg Limit ## Duck Egg Limit	*** Ministry of Health
		<25 km	>25 km	<25 km	>25 km	<25 km	>25 km		
Hg	Mean ± SD	11.93 ± 5.08	17.74 ± 10.07	35.61 ± 16.85	43.90 ± 16.97	60.63 ± 9.42 *	46.30 ± 3.28 *	-	20
	Median	9.82	14.40	35.57	44.03	61.57	45.37		
	Min	6.60	6.90	20.00	22.90	48.97	42.77		
	Max	19.53	33.10	51.20	60.90	71.80	50.93		
Pb	Mean ± SD	44.38 ± 10.44 *	57.03 ± 17.50 *	85.78 ± 19.86	73.25 ± 18.41	66.96 ± 8.33 *	53.52 ± 11.75 *	100	-
	Median	42.97	53.10	78.80	71.88	71.06	55.47		
	Min	32.07	29.85	66.87	52.20	53.18	34.70		
	Max	65.71	102.86	116.50	110.80	77.19	67.29		
Cd	Mean ± SD	12.47 ± 15.01	8.94 ± 5.41	11.33 ± 7.71	12.84 ± 4.71	13.13 ± 13.23	11.92 ± 8.27	-	-
	Median	6.32	7.99	11.19	13.99	6.92	7.45		
	Min	3.66	2.99	4.14	5.55	4.24	5.18		
	Max	65.28	25.46	25.28	20.31	45.11	27.44		
Mn	Mean ± SD	2938.02 ± 741.92	3641.70 ± 1609.26	5178.25 ± 1425.92	4214.80 ± 1162.15	5021.75 ± 1320.39 *	3413.13 ± 759.84 *	-	-
	Median	2704.64	3192.63	5474.80	4615.27	5060.37	3477.37		
	Min	1973.84	1794.86	3258.91	2108.09	3201.24	2259.31		
	Max	4759.35	8432.82	7139.69	5528.35	6920.93	4377.22		

* Significantly different at *p* < 0.05, # National Bureau of Agricultural Commodity and Food Standards Ministry of Agriculture and Cooperatives [68]; ## National Bureau of Agricultural Commodity and Food Standards Ministry of Agriculture and Cooperatives [69]; *** Ministry of Health [70].

**Table 2 foods-11-02791-t002:** Mean ± SD, the median, minimum, and maximum values of Hg, Pb, Cd, and Mn concentrations in poultry blood (µL L^−1^).

Metals		Chicken		Duck Farm		Free-Grazing Duck	
		<25 km	>25 km	<25 km	>25 km	<25 km	>25 km
Hg	Mean ± SD	0.96 ± 0.60	1.33 ± 0.81	2.19 ± 0.91	2.96 ± 1.17	3.07 ± 0.63 *	2.48 ± 0.64 *
	Median	0.69	1.21	1.93	2.98	3.38	2.60
	Min	0.33	0.29	1.37	1.27	2.30	1.58
	Max	2.46	3.07	4.56	4.78	4.57	3.41
Pb	Mean ± SD	22.14 ± 14.85	20.41 ± 13.37	20.96 ± 6.22	26.62 ± 10.92	43.83 ± 20.27	33.08 ± 10.57
	Median	18.70	15.93	20.53	23.83	29.30	23.27
	Min	9.20	7.57	11.03	12.83	13.23	8.40
	Max	77.53	55.97	32.23	44.73	74.10	46.60
Cd	Mean ± SD	2.92 ± 0.99	2.55 ± 0.73	5.45 ± 0.90 *	4.50 ± 0.75 *	5.25 ± 1.17	4.62 ± 0.85
	Median	2.80	2.39	5.84	4.55	5.05	4.33
	Min	1.65	1.50	4.19	3.35	4.31	3.47
	Max	6.18	4.92	6.56	6.08	8.20	6.05
Mn	Mean ± SD	80.51 ± 27.48	72.14 ± 22.97	77.47 ± 20.55	76.31 ± 25.18	55.93 ± 19.66	45.84 ± 13.84
	Median	77.14	72.53	81.54	78.01	55.86	45.87
	Min	33.44	32.47	45.03	42.23	33.15	26.59
	Max	149.18	129.10	118.31	146.56	88.26	68.37

* Significantly different at *p* < 0.05.

Interestingly, this study revealed that there was a correlation between the Hg found in eggs in free-grazing ducks and blood at r^2^ = 0.25 (*p* < 0.05), as shown in Table 3, which is consistent with the report of Heinz et al. (2010), which reported that the concentration of Hg in mallard blood was closely correlated with the concentration of Hg in their eggs (r^2^ = 0.88; *p* < 0.001) [71]. Moreover, there was a correlation between the Pb found in eggs in both chickens and free-grazing ducks and blood at r^2^ = 0.16 and r^2^ = 0.33 (*p* < 0.05), respectively, as shown in Table 3, which is consistent with the report of Trampel et al. (2003), which found that Pb content of the egg yolks strongly correlated with blood Pb levels [72]. Therefore, eggs and blood are considered good bioindicators for monitoring heavy-metal contamination, especially for Hg and Pb [73,74]. In poultry feed, we found no significant difference in heavy-metal concentrations between both areas, as shown in Table 4. On the contrary, the average concentration of Cd in drinking water on chicken farms located in an area <25 km away (0.12 ± 0.05 μL L^−1^) was significantly higher (*p* < 0.05) than for those located >25 km away (0.06 ± 0.03 μL L^−1^), as shown in Table 5.

Farmers interviewed indicated that the water supplied to their animals on the intensive chicken farms located close to the mine (<25 km) came mainly from tap water (66.66%), and 33.33% came from canals and groundwater. For chicken farms located farther away from the mine (>25 km), 83.33% came from tap water and only 16.66% from canals (Appendix A). Torrance et al. (2021) reported that the geochemical data from surface water from streams around gold mining in Colombia were compared to a comprehensive data set of whole-rock analyses from drill-core and channel samples from the deposit, indicating that the deposit is significantly enriched in Pb and Cd compared to crustal averages [75]. Therefore, gold mining may affect Cd contamination in water sources, particularly in the groundwater in this study. Dietary Cd exposure at ≥15 mg kg^−1^ for 6 weeks induced hepatic damage, and increasing dietary Cd concentration increased the residues of Cd in the yolk in laying hens in China [76]. Furthermore, there was a high correlation between the Pb (r^2^ = 0.84) and Cd (r^2^ = 0.42) found in drinking water and blood in free-grazing ducks in an area <25 km away at *p* < 0.05, as shown in Table 6. This is consistent with our previous study, which found a high correlation between Pb concentration in whole eggs and drinking water (r^2^ = 0.806) at *p* < 0.05 for the free-grazing duck farms in Central and Western Thailand [77]. Free-grazing duck flocks raised in an area <25 km away from the gold mine mostly used 100% water (Appendix A). Thus, the canal water may be indicated as a primary source of Pb contamination in the blood of free-grazing ducks.

For the soil, there was a correlation of Pb between the soil on chicken farms in an area <25 km away and eggs at r^2^ = 0.55 (*p* < 0.05), as shown in Table 7. This is consistent with a report by Waegeneers et al. (2009), which found that the Pb concentration in chicken eggs was significantly correlated to the Pb concentration in the soil in the outdoor run (r = 0.49, *p* < 0.001) [78]. Miller et al. (2004) reported significant Cd and Pb contamination of agricultural soils up to 200 km downstream of tin mines in Bolivia, with some concentrations exceeding the recommended guideline values for agricultural use in the Netherlands, Canada, and Germany. These metals flow into the soil, water (including rivers, irrigation canals, and drinking-water supplies), and crops on particular livestock and poultry farms [79].

We also found a correlation in Mn concentrations recorded in soil and blood from chickens between the soil on chicken farms located <25 km away from the mine site (r^2^ = 0.32, *p* < 0.05; Table 7). Hao et al. (2016) reported that the high concentration of Mn was likely due to residual chemicals in the soil after mining activity in China, which had a more significant impact on local water quality than terrace-field farming and poultry-breeding activities [80]. The average concentration of Mn in drinking water on duck farms and free-grazing ducks in both areas was above the water standards for animal consumption by 5–11 times, as shown in Table 5. The 10–100 mg kg^−1^ dosages of Mn can increase apoptosis in young turkeys, increase global DNA methylation, and decrease the activity of antioxidant enzymes [81,82]. Interestingly, the Mn concentration in the feed from chicken and duck farms in this study was found in a range between 57 and 147 mg kg^−1^, which might be a potential risk to poultry health in both areas. There was no correlation found between the feed and eggs in both areas. We found a correlation between Mn levels in the feed and blood of chickens raised on farms located <25 km away (r^2^ = 0.24, *p* < 0.05), as shown in Table 8. This is consistent with the report of Zhao et al. (2019), who reported that the Mn concentrations in the plasma and heart of broilers increased linearly as dietary Mn concentration increased [83]. Furthermore, we also found a significant correlation between Cd in the feed and blood of ducks farmed nearest to the gold mine (r^2^ = 0.95, *p* < 0.05; Table 8). Thus, Cd and Mn concentrations found on the duck and chicken farms <25 km away might be related to the feed used, since the farmers used 50% commercial and 50% semi-commercial feed for duck farms and used 50% commercial and 50% commercial and semi-commercial feed for chicken farms, as shown in Appendix A. From the results of the analysis of heavy metals in animal feed, it was not found that it exceeded the standard limit but should be critically controlled for levels of heavy metals in animal feed and water sources as well as monitored regularly to assess the risks.

#### 3.1.2. Area > 25 km away from the Gold Mine

On the contrary, the average Pb concentration in chicken eggs in an area >25 km away (57.03 ± 17.50 μg kg^−1^ dry weight) was significantly higher than in an area <25 km away (44.38 ± 10.44 μg kg^−1^ dry weight) at *p* < 0.05, as shown in Table 1. Surprisingly, the concentration of Hg and Cd in soil from the chicken farm was also significantly higher than in an area <25 km away, as shown in Table 9. Pb is primarily derived from particular anthropogenic sources, such as traffic, agriculture, and coal burning. Pb exposure occurs through the production and use of Pb-containing products such as Pb gasoline, paint, and Pb pipes in water-distribution systems, indicated to be an important source of potential exposure to general organisms [58,84,85]. Zarcinas et al. (2004) reported that Cd concentrations in soil in Thailand were strongly correlated with organic matter and attributed to the input of contaminants in agricultural fertilizers and soil amendments (e.g., manures, composts) [86]. Moreover, the mobilization of Pb and Cd in soil depends on the persistence of the metal-containing particles in the atmosphere [87]. The location of chicken farms >25 km away was mainly located 100% within the community, with the soil on the farm being dug up and brought back to make manure at 83.33%, whereas in areas <25 km was located within the community at 83.33%, with the soil on the farm dug up and brought back to make manure at 66.66%, as shown in Appendix A. The location and utilization of soil on farms was the main factor causing the Pb and Cd contamination in the >25 km area to be higher than the <25 km area. However, the concentrations of Hg, Pb, Cd, and Mn in both areas did not exceed the standards in soil for residential and agricultural uses, suggesting that the farming areas in Phichit were still safe and suitable for use in agriculture and farming. Hg and Cd contamination in soil and water may cause a significant accumulation in chicken and duck tissues, such as that found in kidneys, liver, and muscles in Spain and China [88,89]. On chicken farms, a correlation was found between Hg concentration in drinking water, eggs (r^2^ = 0.41), and blood (r^2^ = 0.25) at *p* < 0.05. In addition, we found a significant correlation between Pb concentration in drinking water and chicken blood (r^2^ = 0.31; *p* < 0.05), as shown in Table 6. Our study also indicated that Hg and Pb contamination in drinking water may result from tap water since 83.33% of the farmers used it to supply water to their animals (Appendix A). Although the concentration of heavy metals in water on chicken farms located >25 km away from the mine did not exceed the standard limit, monitoring tap-water quality should be carried out regularly to assess the risks.

**Table 4 foods-11-02791-t004:** Mean ± SD, the median, minimum, and maximum values of Hg, Pb, Cd, and Mn concentrations in poultry feed (mgkg^−1^ dry weight).

Metals		Chicken		Duck Farm		Free-Grazing Duck		* Mineral Tolerance of Poultry
		<25 km	>25 km	<25 km	>25 km	<25 km	>25 km	
	Mean ± SD	0.0024 ± 0.0009	0.0027 ± 0.0017	0.0045 ± 0.0028	0.0040 ± 0.0007	-	-	5
Hg	Median	0.0024	0.0021	0.0045	0.0043	-	-	
	Min	0.0013	0.0013	0.0026	0.0031	-	-	
	Max	0.0036	0.0059	0.0065	0.0044	-	-	
	Mean ± SD	0.16 ± 0.09	0.14 ± 0.11	0.18 ± 0.12	0.35 ± 0.38	-	-	10
Pb	Median	0.13	0.12	0.18	0.14	-	-	
	Min	0.08	0.04	0.10	0.12	-	-	
	Max	0.33	0.27	0.27	0.79	-	-	
	Mean ± SD	0.22 ± 0.12	0.14 ± 0.03	0.18 ± 0.03	0.15 ± 0.05	-	-	10
Cd	Median	0.16	0.15	0.18	0.12	-	-	
	Min	0.11	0.08	0.16	0.11	-	-	
	Max	0.44	0.17	0.20	0.20	-	-	
	Mean ± SD	102.74 ± 28.90	114.50 ± 13.96	124.75 ± 15.75	130.04 ± 34.62	-	-	2000
Mn	Median	104.41	113.54	124.75	125.92	-	-	
	Min	57.81	96.90	113.61	96.90	-	-	
	Max	147.62	136.09	135.89	136.09	-	-	

- = No sample, * = Mineral tolerance of poultry [90].

**Table 5 foods-11-02791-t005:** Mean ± SD, the median, minimum, and maximum values of Hg, Pb, Cd, and Mn concentrations in drinking water (µL L^−1^).

Metals		Chicken		Duck Farm		Free-Grazing Duck		Water Standards for Animal Consumption [91,92,93]
		<25 km	>25 km	<25 km	>25 km	<25 km	>25 km	
Hg	Mean ± SD	0.0293 ± 0.0239	0.0176 ± 0.0095	0.0021 ± 0.0018	0.0456 ± 0.0789	0.0356 ± 0.0329	0.0125 ± 0.0042	10
	Median	0.0225	0.0175	0.0021	ND	0.0200	0.0125	
	Min	0.0033	0.0058	0.0008	ND	0.0133	0.0083	
	Max	0.0675	0.0325	0.0033	0.1367	0.0733	0.0167	
Pb	Mean ± SD	1.10 ± 1.30	0.54 ± 0.77	0.05 ± 0.01	0.19 ± 0.08	1.10 ± 1.01	1.14 ± 0.75	100
	Median	0.44	0.11	0.05	0.24	0.54	1.04	
	Min	0.10	0.07	0.04	0.10	0.49	0.44	
	Max	3.36	1.95	0.05	0.24	2.26	1.94	
Cd	Mean ± SD	0.12 ± 0.05 *	0.06 ± 0.03 *	0.21 ± 0.22	0.18 ± 0.13	0.12 ± 0.05	0.72 ± 0.74	50
	Median	0.10	0.07	0.21	0.23	0.15	0.59	
	Min	0.07	0.01	0.06	0.03	0.07	0.05	
	Max	0.21	0.09	0.37	0.28	0.15	1.52	
Mn	Mean ± SD	7.10 ± 5.09	17.04 ± 18.51	287.67 ± 393.37	330.43 ± 315.00	288.67 ± 231.63	560.91 ± 307.24	50
	Median	5.19	10.42	287.67	360.56	188.10	729.99	
	Min	3.72	0.67	9.51	1.45	124.34	206.27	
	Max	17.23	51.17	565.82	629.28	553.59	746.48	

ND = not detected, * significantly different at *p* < 0.05.

**Table 6 foods-11-02791-t006:** Correlations between Hg, Pb, Cd, and Mn in eggs, blood, and drinking water (r^2^-value).

		Metals	Drinking Water									
			<25 km					>25 km				
			Hg	Pb	Cd	Mn	*p*-Value	Hg	Pb	Cd	Mn	*p*-Value
Eggs	Chicken	Hg	0.001				0.9041	0.41 *				0.0040
		Pb		0.01			0.7151		0.001			0.9050
		Cd			0.18		0.0769			0.04		0.4527
		Mn				0.09	0.2293				0.03	0.5100
	Duck farm	Hg	0.10				0.5639	0.02				0.7435
		Pb		0.14			0.4972		0.01			0.7435
		Cd			0.69		0.0583			0.37		0.0857
		Mn				0.04	0.7139				0.16	0.2912
	Free-grazing duck	Hg	0.11				0.3496	0.003				0.8840
		Pb		0.21			0.1808		0.06			0.5292
		Cd			0.06		0.5109			0.03		0.6682
		Mn				0.04	0.5563				0.19	0.2440
Blood	Chicken	Hg	0.06				0.3120	0.25 *				0.0331
		Pb		0.002			0.8571		0.31 *			0.0157
		Cd			0.002		0.8480			0.001		0.8804
		Mn				0.08	0.2610				0.18	0.0762
	Duck farm	Hg	0.01				0.9194	0.08				0.4630
		Pb		0.19			0.4194		0.01			0.8100
		Cd			0.04		0.7139			0.04		0.6134
		Mn				0.01	0.9194				0.001	0.9484
	Free-grazing duck	Hg	0.17				0.2359	0.13				0.3309
		Pb		0.84 *			0.0002		0.003			0.8979
		Cd			0.42 *		0.0443			0.28		0.1392
		Mn				0.01	0.8287				0.08	0.4600

* Significantly different at *p* < 0.05.

**Table 7 foods-11-02791-t007:** Correlations between Hg, Pb, Cd, and Mn in eggs, blood, and soil (r^2^-value).

		Metals	Soil									
			<25 km					>25 km				
			Hg	Pb	Cd	Mn	P-value	Hg	Pb	Cd	Mn	*p*-value
Eggs	Chicken	Hg	0.12				0.1575	0.02				0.5977
		Pb		0.55 *			0.0004		0.02			0.5479
		Cd			0.001		0.8997			0.0003		0.9449
		Mn				0.01	0.7664				0.04	0.4136
	Duck farm	Hg	0.14				0.4972	0.02				0.7435
		Pb		0.01			0.9194		0.02			0.7435
		Cd			0.10		0.5639			0.25		0.1777
		Mn				0.43	0.1750				0.13	0.3363
	Free-grazing duck	Hg	0.004				0.8687	0.08				0.4630
		Pb		0.05			0.5809		0.28			0.1475
		Cd			0.05		0.5809			0.05		0.5809
		Mn				0.11	0.3853				0.003	0.9116
Blood	Chicken	Hg	0.01				0.7507	0.12				0.1659
		Pb		0.02			0.6042		0.05			0.3667
		Cd			0.01		0.7109			0.09		0.2260
		Mn				0.32 *	0.0147				0.04	0.4184
	Duck farm	Hg	0.001				1.0000	0.01				0.8100
		Pb		0.10			0.5639		0.05			0.5517
		Cd			0.02		0.8028			0.12		0.3586
		Mn				0.36	0.2417				0.08	0.4630
	Free-grazing duck	Hg	0.06				0.5206	0.09				0.4366
		Pb		0.23			0.1938		0.004			0.8801
		Cd			0.01		0.8432			0.08		0.4630
		Mn				0.04	0.6134				0.02	0.7435

* Significantly different at *p* < 0.05.

**Table 8 foods-11-02791-t008:** Correlations between THg, Pb, Cd, and Mn in eggs, blood, and feed (r^2^-value).

Metals			Feed									
			<25 km					>25 km				
			Hg	Pb	Cd	Mn	*p*-Value	Hg	Pb	Cd	Mn	*p*-Value
Eggs	Chicken	Hg	0.14				0.1328	0.19				0.0675
		Pb		0.13			0.1370		0.02			0.5416
		Cd			8.6×10^−5^		0.9708			0.10		0.2033
		Mn				0.07	0.2993				0.20	0.0630
	Duck farm	Hg	0.001				1.0000	0.22				0.2125
		Pb		0.29			0.2972		0.19			0.2499
		Cd			0.24		0.3556			0.01		0.7756
		Mn				0.43	0.1750				0.02	0.7081
Blood	Chicken	Hg	0.003				0.8293	0.19				0.0707
		Pb		0.01			0.6625		0.03			0.4616
		Cd			0.06		0.3365			0.05		0.3495
		Mn				0.24 *	0.0399				0.0005	0.9320
	Duck farm	Hg	0.001				1.0000	0.06				0.5517
		Pb		0.001			1.0000		0.06			0.5206
		Cd			0.95 *		0.0010			0.02		0.7081
		Mn				0.07	0.6583				0.13	0.3363

* Significantly different at *p* < 0.05.

**Table 9 foods-11-02791-t009:** Mean ±SD, the median, minimum, and maximum values of Hg, Pb, Cd, and Mn concentrations in soil (mg kg^−1^).

Metals		Chicken		Duck Farm		Free-Grazing Duck		** Soil Standard Limit
		<25 km	>25 km	<25 km	>25 km	<25 km	>25 km	
Hg	Mean ± SD	0.0115 ± 0.0034 *	0.0318 ± 0.0238 *	0.0100 ± 0.0017	0.0207 ± 0.0008	0.0177 ± 0.0007	0.0236 ± 0.0024	22
	Median	0.0105	0.0232	0.0100	0.0206	0.0176	0.0246	
	Min	0.0091	0.0158	0.0087	0.0200	0.0171	0.0209	
	Max	0.0183	0.0792	0.0112	0.0216	0.0185	0.0253	
Pb	Mean ± SD	5.99 ± 2.04	9.75 ± 4.28	6.58 ± 7.58	5.39 ± 2.27	7.41 ± 5.14	14.62 ± 3.31	400
	Median	5.78	10.75	6.58	5.66	10.05	14.57	
	Min	2.62	4.27	1.22	3.00	1.49	11.33	
	Max	8.16	14.12	11.93	7.51	10.70	17.95	
Cd	Mean ± SD	0.16 ± 0.05 *	0.28 ± 0.09 *	0.17 ± 0.09	0.36 ± 0.15	0.11 ± 0.02	0.19 ± 0.08	67
	Median	0.14	0.30	0.17	0.29	0.12	0.21	
	Min	0.11	0.15	0.11	0.24	0.09	0.10	
	Max	0.25	0.41	0.24	0.53	0.12	0.26	
Mn	Mean ± SD	502.19 ± 237.46	509.55 ± 37.81	483.10 ± 192.66	631.64 ± 212.81	304.86 ± 106.95	449.89 ± 175.51	1710
	Median	450.62	523.89	483.10	608.46	340.52	421.19	
	Min	257.34	437.28	346.87	431.37	184.64	290.51	
	Max	859.13	536.73	619.33	855.10	389.43	637.99	

** = Soil-quality standards used for living and agriculture [94], * significantly different at *p* < 0.05.

### 3.2. Carcinogenic Risks

The results showed that the estimated ILCR for both Pb and Cd exceeded the limit set by the USEPA (10^−4^) for all age groups and the two groups of farms tested, being particularly high in the area <25 km away for chicken-egg consumption. The estimated ILCR for Pb and Cd associated with chicken-egg consumption was the highest in the 13–18 yo and 18–35 yo age classes, and the lowest for elders >65 yo (Figure 1A). The ILCR estimated for Pb and Cd associated with duck-egg consumption was the highest for those 18–35 yo and the lowest for elders >65 yo (Figure 1A,B). These results were associated with the fact that the 13–18 and 18–35 yo age groups had the highest consumption of chicken and duck eggs. Pb affects several normal system functions of the human body, and it accumulates in the bones and turns over with a half-life of about 30 years, particularly in the developing nervous systems of fetuses and children [84]. Even at low levels of Pb, children are vulnerable to exposure and suffer irreversible neurological functions, impacting learning, educational attainment, and behavior [95]. In adults, the chronic effects of exposure to Pb include elevated blood pressure, cardiovascular-system damage, neurodegeneration, and development of cancers [96,97]. Both Pb and Cd act as nephrotoxic agents, particularly in the renal cortex [98]. Sohrabi et al. (2018) reported that Pb in cancerous tissues in cases of colorectal cancer was significantly higher than that of healthy tissues (*p* < 0.05), indicating that Pb may play a role in developing colorectal cancer [99]. Chronic Cd exposure may lead to damage to the kidneys, liver, skeletal system, and cardiovascular system, as well as to the deterioration of sight and hearing and the development of cancers of the lung, breast, prostate, pancreas, urinary bladder, and nasopharynx [100,101,102]. O’Brien et al. (2019) reported that positive associations have been reported between urinary Cd concentrations and breast cancer in case-control studies (diagnosis age < 50 years) [103]. Moreover, Cd toxicity can lead to the dual role of inducing liver injury and inhibiting the progression of early liver cancer [104].

Our study revealed an elevated risk of cancer associated with both Pb and Cd consumption, which could have a serious impact on human health, especially for those aged 13–35 yo who consume eggs from an area within 25 km of a gold mine. In 2017, cancer was the most common cause of death in Pichit Province, with a significant increase in cancer death rates from 119.71 in 2015 to 126.3 per 100,000 people in 2017. More specifically, among the population of about 26,155 people living within 25 km of the gold mine, it was reported that 20 people had died of cancer in 2016, including lung cancer, liver cancer, gastrointestinal cancer, heart cancer, and cervical cancer. During the same period, only eight people died of cancer in an area >25 km away (18,288 people), mainly due to liver cancer, cervical cancer, and bladder cancer [105]. However, supporting information and long-term data collection must be carried out to form a robust conclusion of cancer causes.

## 4. Conclusions

The present study revealed that Hg, Pb, and Mn concentrations in eggs from free-grazing ducks on poultry farms located <25 km away from a gold-mine site were significantly higher than on farms located >25 km away from the site. Moreover, Hg in eggs from both farm ducks and free-grazing ducks was 1.5–3 times higher than the standard limit from the Thai Ministry of Public Health. Hg concentrations in the blood of free-grazing ducks raised closer to the gold mine were also significantly higher than in an area >25 km away. Furthermore, the Pb concentrations measured in the blood of farm ducks were also significantly higher on farms located <25 km away and >25 km away. This indicated that free-grazing ducks were exposed to Hg, Pb, and Mn pollution. Despite the traditional free-grazing duck culture in Thailand, thus it is imperative to avoid grazing near gold-mine sites.

Surprisingly, Pb concentrations measured in chicken eggs and the Hg and Cd concentrations in soil from chickens on poultry farms were significantly higher for the samples collected in an area >25 km away. This might be a point to be evaluated in further studies concerning the relation of chemical uses in agriculture and rice cultivation or any other activities in areas far from a gold-mining source. Moreover, the estimated ILCR for both Pb and Cd exceeded the cancer limits (1 × 10^−4)^ for all age groups in both areas and was particularly high in the area <25 km away for chicken-egg consumption, especially among people aged 13–18 and 18–35 years old. Thus, these findings indicate that effective measures to prevent heavy-metal contamination of humans and animals from mining sites are needed, even years after mining operations have stopped. It is particularly important to set regular surveillance and implementation of contingency plans for pollution control and measurement. Recommendations and regular monitoring should be carried out in other livestock production and related food production near gold-mining areas. The bioaccumulation of certain pollutants must be managed regularly in the near future, even after a gold mine starts operations.

## Figures and Tables

**Figure 1 foods-11-02791-f001:**
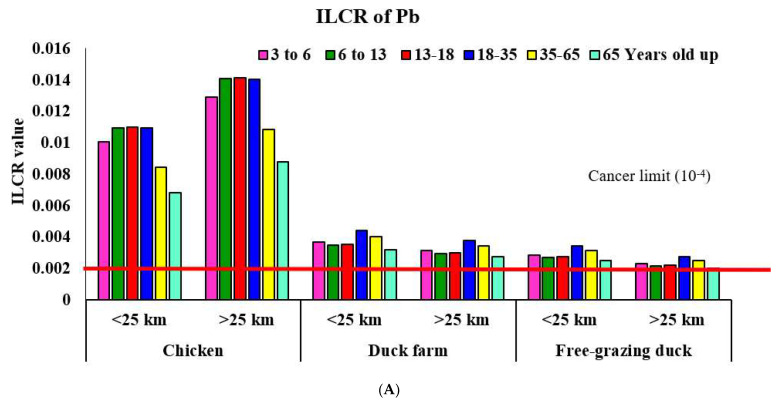
ILCR (incremental lifetime cancer risk) of Pb (**A** = upper graph) and Cd (**B** = lower graph) from egg consumption in areas located <25 km and >25 km from the gold-mine site (the red line indicates a cancer limit of 10^−4^ set by the USEPA).

**Table 3 foods-11-02791-t003:** Correlations between Hg, Pb, Cd, and Mn in eggs and blood (r^2^-value).

		Blood					*p*-Value
			Hg	Pb	Cd	Mn	
		Hg	0.005				0.6788
	Chicken	Pb		0.16 *			0.0169
		Cd			0.06		0.1449
		Mn				5 × 10^−^^5^	0.9667
Eggs		Hg	0.001				0.8994
	Duck farm	Pb		0.07			0.3344
		Cd			0.14		0.1728
		Mn				0.070	0.3412
		Hg	0.25 *				0.0361
	Free-grazing duck	Pb		0.33 *			0.0122
		Cd			0.001		0.8933
		Mn				0.13	0.1421

* Significantly different at *p* < 0.05.

## Data Availability

Data are contained within the article.

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
