# Peer review of "Potential Health Effects of Heavy Metals and Carcinogenic Health Risk Estimation of Pb and Cd Contaminated Eggs from a Closed Gold Mine Area in Northern Thailand"

_foods, 2022, doi:10.3390/foods11182791_

Round 1
Reviewer 1 Report
Comments:
1. Check “cº”.
2. Use one format for units throughout the manuscript. I mean using “/” or “-1”.
3. How metals have find their way to farms? by water? by air?
4. Remove word “Axis Title” from Figure 1.
5. Why unit of metals in eggs is mentioned as “µg/L” in Table 1? Why not mentioned as “mg/kg or ug/kg”?
6. Use complete form of “THg” when used first time in the manuscript.
Author Response
Response to Reviewers
Dear reviewers
Authors are thankful to all reviewers. Their comments will definitely help us to improve the quality of the manuscript. Details of corrections are listed below point by point.
Reviewer 1
- Check “cº”.
Answer Author edited according to the comment as shown in Line no. 142 and 152.
- Use one format for units throughout the manuscript. I mean using “/” or “-1”.
Answer Author edited according to the comment in both whole manuscript and Table.
- How metals have find their way to farms? by water? by air?
Answer: In this study, authors suspected them from water resource. However, we also determined them in feed, soil as well. Therefore, from our finding the main route of heavy metal contamination come from feed and water. Unfortunately, authors did not determine in the air, which it might be another route of contamination and further study needed to be done.
- Remove word “Axis Title” from Figure 1.
Answer Author edited according to the comment as shown in Figure1.
- Why unit of metals in eggs is mentioned as “µg/L” in Table 1? Why not mentioned as “mg/kg or ug/kg”?
Answer Author edited it becomes “µg kg-1” as shown in Table 1.
- Use complete form of “THg” when used first time in the manuscript.
Answer Author edited according to the comment by using “Hg” as shown in manuscripts. This correction was similar to the suggestion from reviewer no. 3.
Best regards,
Phitsanu Tulayakul
Reviewer 2 Report
Dear Editor and Authors
The study presents an adequate approach to assess a problem that is relevant at the local level, but of public health interest.
Minor comments
Line 60: Hg has been a priority pollutant for some decades and the concern with exposure to Hg is not new.
Lines 63-64: According to IARC, Group 2B (Possibly carcinogenic to humans) and Group 1 (Carcinogenic to humans).
2.3. Analytical procedure: please, include the mean and coefficient of variation of recoveries.
Tables: review signs greater-than and lesser-than.
Author Response
Response to Reviewers
Dear reviewers
Authors are thankful to all reviewers. Their comments will definitely help us to improve the quality of the manuscript. Details of corrections are listed below point by point.
Reviewer 2
Dear Editor and Authors
The study presents an adequate approach to assess a problem that is relevant at the local level, but of public health interest.
Minor comments
Line 60: Hg has been a priority pollutant for some decades and the concern with exposure to Hg is not new.
Answer Author agreed with the comment, however, the study of heavy metal related food safety is limited in Thailand. Therefore, it would be nice to have many study report on this finding as the witness of situation in this region.
Lines 63-64: According to IARC, Group 2B (Possibly carcinogenic to humans) and Group 1 (Carcinogenic to humans).
Answer Author edited best according to the comment as shown in line no. 68-70.
2.3. Analytical procedure: please, include the mean and coefficient of variation of recoveries.
Answer Author edited best according to the comment as shown in line no.162.
2.4.Tables: review signs greater-than and lesser-than.
Answer Author edited according to the comment as shown in the table.
Best regards,
Phitsanu Tulayakul

Reviewer 3 Report
1、The title is “… contaminated eggs …”. The study focused on heavy metal contamination in the eggs, blood, feed, soil, and drinking water on chicken farms, …, and the title is “… Pb and Cd…”, Hg and Mn were also found in the manuscript, so it's best to recap the title.
2、“Hg” and “THg”, it's better to write them in the same form.
3、In section 2.2, line139, the number of parentheses does not match, maybe missing the other half of the parentheses.
Author Response
Response to Reviewers
Dear reviewers
Authors are thankful to all reviewers. Their comments will definitely help us to improve the quality of the manuscript. Details of corrections are listed below point by point.
Reviewer 3
1、The title is “… contaminated eggs …”. The study focused on heavy metal contamination in the eggs, blood, feed, soil, and drinking water on chicken farms, …, and the title is “… Pb and Cd…”, Hg and Mn were also found in the manuscript, so it's best to recap the title.
Answer Thank you for your comment and author would like to remain the same title which broaden and cover all necessary information.
2、“Hg” and “THg”, it's better to write them in the same form.
Answer Author edited according to the comment by using “Hg” in the whole manuscripts.
3、In section 2.2, line139, the number of parentheses does not match, maybe missing the other half of the parentheses.
Answer Author edited according to the comments by adding “)”as shown in Line no. 144.
Best regards,
Phitsanu Tulayakul
